# Flow Measurement of Oil-Water Two-Phase Flow at Low Flow Rate Using the Plug-in Conductance Sensor Array

**DOI:** 10.3390/s19214649

**Published:** 2019-10-25

**Authors:** Ningde Jin, Yiyu Zhou, Xinghe Liang, Dayang Wang, Lusheng Zhai, Jidong Wei

**Affiliations:** School of Electrical and Information Engineering, Tianjin University, Tianjin 300072, China; ndjin@tju.edu.cn (N.J.); zhouyiyu@tju.edu.cn (Y.Z.); lxh19881020@yahoo.com.cn (X.L.); wangdayang@tju.edu.cn (D.W.); lszhai@tju.edu.cn (L.Z.)

**Keywords:** oil-water two-phase flow at low flow rate, plug-in conductance sensor array, water holdup, water cut

## Abstract

In order to improve the flow measurement accuracy of oil-water two-phase flow at low flow rate, this paper presents a plug-in conductance sensor array (PICSA) for the measurement of water holdup and cross-correlation velocity. Due to the existence of the insert body in PICSA, the effect of slippage and the non-uniform distribution of dispersed phase on the measurement of oil-water two-phase flow at low flow rate can be reduced. The finite element method is used to analyze the electric field distribution characteristics of the plug-in conductance sensor, and the sensor geometry is optimized. The dynamic experiment of oil-water two-phase flow is carried out where water cut *K*_w_ and mixture velocity *U*_m_ are set in the range of 10–98% and 0.0184–0.2580 m/s respectively. Experimental results show that the PICSA has good resolution in water holdup measurement for dispersed oil-in-water slug flow (D OS/W), transition flow (TF), dispersed oil-in-water bubble flow (D O/W) and very fine dispersed oil-in-water bubble flow (VFD O/W). In addition, the cross-correlation velocity of the oil-water two-phase flow is obtained by using the plug-in upstream and downstream conductance sensor arrays. The relationship between the cross-correlation velocity and mixture velocity is found to be sensitive to the change of flow pattern, but it has a good linear relationship under the same flow pattern. Based on the flow pattern identification, a good prediction result of the mixture velocity is obtained using kinematic wave theory. Finally, a high precision prediction of the individual phase volume fraction of oil-water two-phase flow at low flow rate is achieved by using the drift flux model.

## 1. Introduction

Due to the long-term exploitation for oil by injecting water, the production of oil reservoirs with low permeability and low fluid productivity is characterized by oil-water two-phase flow at low flow rate and high water-cut. Accurate prediction of flow parameters such as mixture velocity and water holdup are of great significance for optimizing oilfield mining schemes. At present, the water cut of old oil fields is generally over 80% in China. For example, at the end of 2007, the comprehensive average water cut of all new and mature oil fields owned by the three major oil companies in China has reached 86.0%. Generally, when the water cut exceeds 80%, oilfields are regarded to enter the late stage of high water-cut development [1]. When the flowrate is less than 0.2580 m/s, the slippage effect between phases is extremely seriously and the distribution of dispersed phase is extremely non-uniform than those at high flow rate conditions. These flow conditions are defined as low flow rate conditions. Under the low flow rate and high water-cut conditions, the flow measurement is a great challenge, and it is urgently needed to be investigated. For the measurement of mixture velocity, single phase flow meters are often used [2,3]. But they are susceptible to mixed fluid concentrations and flow patterns, which limit their application. Cross-correlation method is an online flow parameter detection technology based on stochastic process theory and information theory [4,5,6]. The measurement accuracy of mixture velocity depends on the validity of the relationship between cross-correlation velocity and mixture velocity [7]. The study found that the cross-correlation velocity measured by the cross-correlation method corresponds to the velocity of kinematic wave in the multiphase flow [8,9]. Based on the equivalence between cross-correlation velocity and kinematic wave velocity, Lucas and Walton [10] use the cross-correlation measurement method to measure the total flow velocity of oil-water two-phase flow. Lucas and Jin [11] subsequently corrected the correlation flow rate measurement model of oil-water two-phase flow based on the kinematic wave theory and obtained good measurement results. The cross-correlation method has the advantages of simple implementation, continuous measurement, good applicability and so on.

In terms of water holdup measurement, the conductance method is widely used due to its simple sensor structure and fast response [12,13,14,15,16]. The electrode structure of the conductance sensor affects the electric field distribution, which in turn affects the measurement of flow parameters. Coney [17] earliest used a flat-conductance sensor to measure the thickness of a liquid film in a gas-liquid two-phase flow. However, the plate electrode cannot be completely fitted with the circular pipe, and the resolution is lower than that of the ring-shape electrode [18,19]. Asali et al. [20] measured the two-phase flow in a vertical pipe using a ring-shape conductance sensor. Tsochatzidis et al. [21] used a set of conductance rings to identify the flow pattern in the pipe. According to the measurement results, the ring-shape conductance sensor can be used not only to identify the flow pattern, but also to measure the average phase fraction. Liu et al. [22] used a vertical multiple electrode array conductance sensor to measure the water holdup of oil-water two-phase flow with different salinities. Han et al. [23] investigated the holdup phenomena of low velocity oil-water flows in a vertical upward small diameter pipe based on the measurement results of the multiple conductance sensor array. However, Jin et al. [24] found that the electric field of the ring-shape electrode is concentrated around the pipe wall and is not sensitive to the change of water holdup at the center of the pipe. Costigan et al. [25] designed an arc-type conductance sensor to measure the gas holdup, and found that the output signal of the sensor has a good linear relationship with the phase fraction. Kytöma et al. [9] installed a protective electrode at each end of the curved electrode to make the electric field distribution more concentrated, suppressing the edge effect of the electrodes and making the sensitive field distribution more uniform. Since then, the method of adding guard electrodes has been widely used to improve the measurement sensitivity of conductance sensors [26,27,28]. Han et al. [29] demonstrated that sensor with arc-type electrodes presented higher resolution in water holdup measurement compared to ring-shape ones. However, the arc-type conductance sensor is susceptible to flow patterns and has limitations in the measurement of two-phase flow parameters with severe slippage effect. Gao et al. [30] designed a new four-sector distributed conductance sensor for multi-channel measurement. Experiments show that the measurement resolution of the sensor needs to be further improved. Wei et al. [31] designed a three-channel conductance probe with center body. The water-holdup measurement characteristics were improved by inserting the center body to accelerate fluid flow rate, but there still exist the same measurement limitations which traditional arc-type conductance sensors have. The electrical resistivity tomography (ERT) has the advantages of non-intrusive, online measurement, etc. Pedersen et al. [32] used ERT method to detect 3-phase slug flow where the oil-to-water ratio is low. Durdevic et al. [33] evaluated different algorithms for creating 2-D images and the feasibility of estimating oil in water concentrations using ERT method, and for high water cut, the overall content of the pipe can still be determined. 

In view of the uneven electric field distribution and low resolution of the conductance sensor with ring-shape electrode and the arc-type conductance sensor is easily affected by slippage effect and the non-uniform distribution of the flow structure, this paper designs a plug-in conductance sensor array used in the low-flow-rate oil-water two-phase flow with extremely severe slippage effect and uneven distribution of dispersed phase. It can reduce the slippage effect and the non-uniform distribution of the dispersed phase by focusing the fluid to flow through the annular space and increase the fluid velocity. The finite element method is used to obtain the optimal sensor geometry. The dynamic experiment is carried out to evaluate the sensor performance. The models are established to predict the mixture velocity and the individual phase volume fraction of oil-water flows.

## 2. Geometry Optimization of the PICSA

The structural dimensions of the plug-in conductance sensor are shown in Figure 1. It consists of an insert body, two support frames, and four ring-shape electrodes. Two ring-shape electrodes form a sensor, one is the exciting electrode and the other is the measuring electrode. A single sensor is used for the measurement of water holdup, and the combination of the upstream and downstream sensors are used for the acquisition of cross-correlation velocity. The fixture is a pair of stainless steel cylinders that pass through the pipe and insert body to ensure that the insert body is stable during fluid flowing. The inner diameter of the pipe *D_d_* = 20 mm. The diameter of the insert body *D* = 10 mm, the length *L* = 93 mm, and the radius of the arc-shape area at both ends *R_a_* = 5 mm. Stainless steel cylinder diameter *R_d_* = 3 mm. As the electrodes are flush mounted on the insert body, the outer diameter of the conductance ring *R_eo_* = 10 mm, which is the same as the diameter of the insert body. The inner diameter of the electrodes *R_ei_* = 8 mm. The parameters to be optimized are the height of the electrode *h*, the distance between the electrodes *H*, and the distance between the upstream and downstream sensors *d*. The purposes of the optimization are that each sensor has a strong and uniform detection field in the measurement area, and the detection fields between the upstream and downstream sensors do not interfere with each other.

The sensors are optimized using the finite element analysis software ANSYS. In order to describe the sensitivity distribution characteristics of the detection field quantitatively, the current *I* = 0.1 mA is applied to the exciting electrode, and the measuring electrode is connected to the ground with current −*I* = −0.1 mA exerted on it. Changing the electrode height *h* and the distance between the electrodes *H*, the sensitivity distribution characteristics of the detection field are investigated according to the change of the potential on the exciting electrode. The so-called detection field sensitivity refers to the response of each position in the measurement area of the sensor to the change of the nature of the medium (in this case, the medium is alternating oil and water). The electric field distribution can be described with Laplace equation $\nabla^{2}U\left(x,y,z\right)=0$. The boundary conditions of the sensor can be expressed as:(1)$$\left\{\begin{array}{l} u=0,\;\varphi \in [0,2\pi ],r=\frac{D_{}}{2},z\in [\frac{H-h}{2},\frac{H+h}{2}]\\ \frac{\partial u}{\partial r}=\frac{I}{Se},\;\varphi \in [0,2\pi ],r=\frac{D_{}}{2},z\in [-\frac{H+h}{2},-\frac{H-h}{2}]\\ \frac{\partial u}{\partial z}=0,\;z=\pm \frac{L_{d}}{2}; \end{array}\right.,$$
where *u* is voltage, and *r*^2^
*= x*^2^
*+ y*^2^*. I* represents the value of exciting current and *Se* refers to the superficial area of exciting electrode. *H*, *h* and *L_d_* correspond to the distance between the electrodes, the electrode height, as well as the axial length of the fluid along the pipe in the finite element calculation model, respectively.

In order to obtain the sensitivity of the detection field, the potential difference *U*_0_ between the two electrodes is measured when the fluid is single water phase. Then place a small non-conductive ball in a position in the electric field distribution area to simulate oil droplet, and the potential difference is *U(x, y)*. The sensitivity distribution can be obtained by traversing all positions in the electric field. The sensitivity can be expressed as [31]:(2)$$S(x,y)=\frac{\Delta U\left(x,y\right)}{{\left[\Delta U\left(x,y\right)\right]}_{\max}}\times 100\%,$$
where $\Delta U\left(x,y\right)=U\left(x,y\right)-U_{0}$, and ${\left[\Delta U\left(x,y\right)\right]}_{\max}$ is the maximum.

In this paper, the average sensitivity (*S_avg_*) and the sensitivity variation parameter (*S_VP_*) have been introduced as the evaluation function of the sensitivity field. The average sensitivity (*S_avg_*) means the average value of the relative sensitivity of each position in the detection field
(3)$$S_{avg}=\frac{1}{M}\sum_{j=1}^{M}S(x,y).$$

According to the concepts of standard deviation and rate of change in statistics, the sensitivity variation parameter is defined as:(4)$$S_{vp}=\frac{S_{dev}}{S_{avg}}\times 100\%,$$
where *S_dev_* represents the deviation of the sensitivity. It can be expressed as:(5)$$S_{dev}={\left(\frac{1}{M}\sum_{j=1}^{M}{\left(S(x,y)-S_{avg}\right)}^{2}\right)}^{1/2}.$$

It can be seen that the *S_avg_* is larger, the more sensitive the sensor is. The *S_vp_* is smaller, the more uniform the sensitivity of the detection field is, which means that the measurement consistency of the sensor is better. Figure 2 shows the *S_avg_* and *S_vp_* of the sensitive fields under different sensor configurations.

It can be seen that the sensitive field is the best when the distance *H* = 7 mm and the height of the electrodes *h* = 2 mm. The sensitive field distribution in this case is shown in Figure 3.

Then, the distance *d* between the upstream and downstream sensors is optimized, so that the electric field between the upstream and downstream sensors does not interfere with each other and the upstream and downstream signals have a good correlation. In the simulation, the exciting electrodes are excited by 5 V voltage, and the measuring electrodes are connected to the ground. Changing the distance *d* between the upstream and downstream sensors, the electric fields do not interfere with each other when *d* is set as 20 mm. The electric field distribution is shown in Figure 4.

## 3. Experiment of Oil-Water Two-Phase Flow at Low Flow Rate and High Water-Cut

The experimental setup is shown in Figure 5. The test section is an acrylic material pipe with 20 mm inner diameter. In order to acquire the fully developed flow patterns in the pipe, a high-speed camera is installed at the height of 2000 mm from the entrance to record the flow structures. Afterward, the plug-in conductance sensor array is mounted at the height of 400 mm from the camera. So that the single sensor measures the water holdup, and the two upstream and downstream sensors measure cross-correlation velocity. In addition, experimental flow mediums include tap water and No. 3 industrial white oil stored in water and oil tank respectively and separated in the mixing tank. Two industrial peristaltic pumps are used to transport and control the flow of water and oil, respectively, with an uncertainty of ±0.2% [34]. The peristaltic pump working based on the discrepancies of rotating speed should be calibrated before the experiment when fluid mediums have distinctive physical properties, such as density and viscosity. Specific calibration procedure can refer to our previous study [34]. 

In the experimental process, water-cut *K_w_* is firstly fixed and with the increasing mixture velocity *U_m_*, both the DC and AC signals of the plug-in conductance sensor array are collected after the flow patterns are fully developed. The analog sinusoidal signals with 20 kHz frequency and peak to peak value of 4 V are applied to the exciting electrodes through reference resistant *R*_ref_, then the measuring electrodes are connected to the ground. The voltages in the sensors and reference resistant *R*_ref_ are demodulated and the outputs of conductance sensors are collected with PXI4472 synchronous acquisition card produced by NI Company. According to the research results reported by Han et al. [34], the highest flowing frequency of low-velocity oil–water flows is about 50 Hz. Herein, 4 kHz is selected as suitable sampling frequency. The sampling time for each flow condition is 30 s, when the flow rate or oil-cut is very low, the sampling time is set as 2 min. When all the mixture velocities are implemented, change water-cut to the next value and repeat the procedure of increasing mixture velocity. Water-cut *K_w_* and mixture velocity *U_m_* are set in the range of 10–98% and 0.0184~0.2580 m/s respectively.

The high-speed camera is used to record the flow pattern information in the experiment, and dispersed oil-in-water slug flow (D OS/W), dispersed oil-in-water flow (D O/W), very fine dispersed oil-in-water flow (VFD O/W) and transition flow (TF) are observed in this experiment. As shown in Figure 6a, oil-in-water slug flow occurs when the total flow rate of oil and water is low, and the turbulent energy of the mixed fluid is small, which is not enough to break the oil phase into a smaller dispersed phase. Due to the low mixture velocity, the probability of occurrence of large oil bubbles in the pipeline increases, and the large oil bubbles are subjected to greater resistance in the pipe, leading to a low rising velocity. The small oil bubbles after the large oil bubbles rise fast because of their small volume. Therefore, small oil bubbles will appear in the pipe to catch up with large oil bubbles in front, and then the collision and coalescence of these oil bubbles will result in the formation of oil slug. For dispersed oil-in-water flow in Figure 6b, with the increase of mixture velocity, the turbulent energy of the mixed fluid gradually increases, and the oil slugs are broken into large oil bubbles. Due to similar velocity of these oil bubbles, the coalescence phenomenon is reduced. The oil phase is present in the form of dispersed oil bubbles in the continuous water phase. For the very fine dispersed oil-in-water flow in Figure 6c, when mixture velocity further increases, the turbulent energy is increased enough to break the oil bubbles into very fine ones which are evenly distributed. Under the very fine dispersed oil-in-water flow, the macroscopic flow structure of the mixed fluid is stable, and the size of the oil bubbles is uniform. For the transitional flow in Figure 6d, as the total flow rate increases, the flow pattern begins to transition from the slug flow to the bubble flow. In this transitional process, large water droplets begin to appear in the continuous oil phase, and occasionally the continuous oil phase is cut off and the water phase present continuous. The flow characteristics are complex and abnormal.

## 4. Water Holdup Measurement Characteristics of the PICSA

When the conductance sensor measures water holdup of the oil-water mixed fluid, the conductivity of the mixed fluid and the holdup of the dispersed phase satisfy the Maxwell [35] relationship:(6)$$\frac{\sigma_{m}}{\sigma_{w}}=\frac{2(1-Y_{s})}{2+Y_{s}},$$
where *Y_s_* are holdup of dispersed phase, *σ_m_* and *σ_w_* are the conductivity of the oil-water mixture and water respectively. As the water holdup *Y_w_* = 1 − *Y_s_*, then the relationship between water holdup and conductivity is:(7)$$\frac{\sigma_{m}}{\sigma_{w}}=\frac{2Y_{w}}{3-Y_{w}}.$$

Define the normalized conductivity $G_{e}^{*}$ as
(8)$$G_{e}^{*}=\frac{\sigma_{m}}{\sigma_{w}}=\frac{V_{\mathrm{ref}}/V_{\mathrm{sen}}}{V_{\mathrm{ref}}^{w}/V_{\mathrm{sen}}^{w}},$$
where *V*_ref_ and *V*_sen_ are the sensor outputs under the circumstance of oil-in-water flows. $V_{\mathrm{ref}}^{w}$ and $V_{\mathrm{sen}}^{w}$ denote the sensor outputs under the circumstance of water.

From Equations (7) and (8), the water holdup can be determined:(9)$$Y_{w}=\frac{3G_{e}^{*}}{2+G_{e}^{*}}.$$

At lower flow rate, the signals of the plug-in conductance sensor are shown in Figure 7a. Since the mixture velocity of the fluid is low at this time, the flow pattern is D OS/W. When the oil slug flows through the sensor sensitive area, the conductance value is significantly reduced, so the normalized conductivity value measured by the sensor drops significantly. When the water cut is low, the number of oil slugs is relatively large, and the length is relatively long. At this time, the amplitude of the sensor voltage signal is relatively large, and the duration is relatively long. As the water cut increases, the size of the oil slug gradually decreases, and the time passing through the sensor sensitive field also becomes shorter. At this time, the amplitude of the downward fluctuation of the measured value of the sensor gradually decreases, and the duration decreases. In addition, the fluid between the two oil slugs is basically water, so the base value of the normalized conductivity measured by the sensor fluctuates around 1.

When the flow rate increases, the signals of the plug-in conductance sensor are shown in Figure 7b. The flow pattern is converted to D O/W. At this time, the oil bubbles of different sizes are evenly distributed in the continuous water phase and continuously pass through the sensitive field of the sensor. The base value of the normalized conductivity value measured by the sensor does not fluctuate around 1, and the volatility decreases. As the water cut increases gradually, it has a relatively clear resolution.

At a higher flow rate, the measurement results of the plug-in conductance sensor are shown in Figure 7c. The flow pattern is VFD O/W, and the oil phase is mainly present in the form of fine oil droplets in water, and its distribution is more uniform. At this time, the sensor’s measured normalized conductivity value is more stable and the fluctuation is smaller. The resolution of the sensor to measure the normalized conductivity value is also satisfactory.

According to the measurement signal of the plug-in conductance sensor, after taking the average value, the measurement chart of water holdup in oil-water two-phase flow at low flow rate and high water-cut is established in Figure 8. The abscissa is set as the total flow of the mixed fluid, and the ordinate is set as the measured value of the water holdup. It can be found that due to the low flow rate, the oil-water phase slippage is severe, and the measured value of the sensor gradually decreases as the total flow rate increases. When the total flow rate is higher than 0.1106 m/s, the water holdup measurement has better step performance and the resolution is obvious. When the total flow rate is less than 0.0737 m/s, the step of the water holdup measurement value becomes small, which is caused by the slippage effect of the oil and water. When water cut is less than 40%, the oil phase is continuous, and the conductive path of the plug-in conductance sensor is cut off, and normal conductance measurement cannot be performed.

## 5. Cross-Correlation Velocity Measurement 

The cross-correlation velocity measurement theory was proposed by Beck [4] in 1981. Two identical sensors are installed on the pipeline, and the output signals are set to *x(t)* and *y(t)* respectively. When the oil-water two-phase flow flows in the pipeline, the flow “noise” related to the flow state is generated during the flow processing. At a certain distance, it can be assumed that the flow of the fluid in the pipe is smooth and random, that is, the “noise” signal of the measured fluid is stable. It can be seen that the signals *x(t)* and *y(t)* are similar, but there is a time delay *τ*_0_, namely:(10)$$x(t)=y(t+t_{0}),$$
where *τ*_0_ is the time that the fluid flows from the upstream sensor to the downstream sensor, also known as the transit time, which corresponds to the flow velocity of the fluid.

According to the cross-correlation theory, the correlation functions of *x(t)* and *y(t)* can be obtained by the following equation:(11)$$R_{xy}\left(\tau \right)=\underset{T\to \infty }{\lim}\frac{1}{T}\int_{0}^{T}x\left(t\right)y\left(t+\tau \right)dt.$$

The cross-correlation function *R_xy_(τ)* reflects the degree of correlation between the two signals *x(t)* and *y(t)*. The time *τ*_0_ corresponds to the maximum value of *R_xy_(τ)*. According to the transit time *τ*_0_ and the distance *d* between the upstream and downstream sensors, the cross-correlation velocity *U_cc_* of the fluid can be obtained:(12)$$U_{cc}=d/\tau_{0}.$$

In this paper, the cross-correlation velocity measurement is performed according to the signals of upstream sensor and downstream sensor of the PICSA under different flow conditions. For D OS/W, the sensor signals are shown in Figure 9a. When a large oil slug flows through the sensor, the sensor produces large fluctuations in signal amplitude, but the oil slug flows through the sensor at a low frequency. Due to the presence of large oil slugs, the stability of fluid flow is good. It can be seen from the figure that the signals of the two sensors are similar, and the correlation peaks of the correlation functions are obvious. For D O/W, the sensor signals are shown in Figure 9b. Although they are not as stable as the large oil slug fluid structure, due to the presence of large oil bubbles in the fluid, the signals of the upstream and downstream sensors are very similar. The correlation is still very good, and the correlation peaks of the correlation functions are obvious. For VFD O/W, the sensor signals are shown in Figure 9c. The amplitude of the signal is significantly smaller than the above two flow patterns, because the very fine oil bubbles are dispersed in the water, resulting in less fluctuation of the conductivity of the mixed fluid. When the small oil bubbles flow through the pipeline, the upstream and downstream sensors can still acquire their fluctuation signals, and the upstream and downstream signals are similar. So, the peak value of the correlation function is still obvious, and the transit time can be accurately obtained.

The mixture velocity *U_m_* can be predicted based on the cross-correlation velocity *U_cc_*. Kytömaa and Brennen [9] considers that the cross-correlation velocity *U_cc_* corresponds to the kinematic wave velocity *U_kw_* in the two-phase flow.
(13)$$U_{cc}=U_{kw}.$$

The kinematic wave velocity *U_kw_* can be expressed as (Zuber [36]):(14)$$U_{kw}=\frac{\partial U_{sd}}{\partial y_{d}},$$
where the superficial velocity of the dispersed phase *U_sd_* can be expressed as (Zuber and Findlay [37]):(15)$$U_{sd}=y_{d}[C_{0}U_{m}+U_{\infty }{\left(1-y_{d}\right)}^{n}],$$
where *C*_0_ is the phase distribution coefficient, and *U_∞_* is the terminal rising velocity of a single oil droplet in infinite still water. The Equation (14) is substituted into (15) to obtain the kinematic wave velocity *U_kw_*:(16)$$U_{kw}=U_{m}\left[C_{0}(U_{m},y_{d})+y_{d}\frac{\partial C_{0}(U_{m},y_{d})}{y_{d}}\right]+U_{\infty }{\left(1-y_{d}\right)}^{n}\left[1-n\frac{y_{d}}{1-y_{d}}\right].$$

Make:(17)$$C_{0}^{*}=C_{0}(U_{m},y_{d})+y_{d}\frac{\partial C_{0}(U_{m},y_{d})}{y_{d}},$$
(18)$$B^{*}=U_{\infty }{\left(1-y_{d}\right)}^{n}\left[1-n\frac{y_{d}}{1-y_{d}}\right],$$

Then the cross-correlation velocity *U_cc_* can be expressed as:(19)$$U_{cc}=U_{kw}=C_{0}^{*}U_{m}+B^{*}.$$

It can be seen that the relationship between *U_m_* and *U_cc_* is related to the values of *C*_0_*** and *B**. The relationship between *U_m_* and *U_cc_* measured in the experiment is shown in Figure 10. The mixture velocity *U_m_* under same flow pattern has a good linear growth trend with the increase of the cross-correlation velocity *U_cc_*. This is because the annular space formed by the plug-in conductance sensor and the inner wall of the pipe can increase the fluid velocity, so that the phase distribution and the slippage effect tend to be consistent. Thus, when the flow patterns are the same, *C*_0_*** and *B** tend to be constant values, respectively. Using the statistical model given in Figure 11, the mixture velocity *U_m_* can be predicted from the measured *U_cc_*. Figure 11 shows the predicted results for mixture velocity *U_m_**^pre^*, where the absolute average percentage deviation (AAPD) is 8.5%.

## 6. Phase Volume Fraction Prediction

Due to the differences in slippage effect, phase distribution and dispersed phase size in different flow patterns, we established a drift flux model (Zuber and Findlay [37]) based on flow patterns to obtain the water cut and oil cut. The drift flux model is expressed as follows:(20)$$\frac{U_{so}}{Y_{o}}=C_{0}U_{m}+\frac{J_{om}}{Y_{o}},$$
where *U_so_* is the superficial velocity of the oil phase, *Y_o_* is the oil holdup, and *C*_0_ is the phase distribution coefficient. *J_om_* is the drift flux and it can be expressed as:(21)$$J_{om}=U_{\infty }Y_{o}{\left(1-Y_{o}\right)}^{n}.$$

A more detailed expression of the drift model can be obtained by combining Equations (20) and (21):(22)$$\frac{U_{so}}{Y_{o}}=C_{0}U_{m}+U_{\infty }{\left(1-Y_{o}\right)}^{n},$$
where three parameters C_0_, n and $U_{\infty }$ need to be determined. First, by changing the bubble diameter index n value, the distribution of ($U_{m}/{\left(1-Y_{o}\right)}^{n}$, $U_{so}/Y_{o}{\left(1-Y_{o}\right)}^{n}$) in the two-dimensional plane is examined, as shown in Figure 12. For D OS/W and TF, when n is equal to 2.7, the data points have the best linear relationship, and the values of C_0_ and are determined as 1.10165 and 0.02998 m/s by linear fitting method, respectively. When n is equal to 2.4, the data points of D O/W and VFD O/W have the best linear relationship, and the values of C_0_ and are determined by linear fitting method to be 0.91257 and 0.0256 m/s, respectively. Therefore, the final expression of the drift flux model based on flow patterns can be expressed as:(23)$$\left\{\begin{array}{ll} \frac{U_{so}}{Y_{o}}=1.10165U_{m}+0.02998{\left(1-Y_{o}\right)}^{2.7} & \left(D\;\mathrm{OS}/W,\;\mathrm{Transition}\;\mathrm{Flow}\right)\\ \frac{U_{so}}{Y_{o}}=0.91257U_{m}+0.0256{\left(1-Y_{o}\right)}^{2.4} & \left(D\;O/W,\;\mathrm{VFD}\;O/W\right) \end{array}\right..$$

Since water cut *K_w_* = (1 − *U_so_*)/*U_m_*, by substituting the mixture velocity prediction model into Equation (23), the prediction of the water cut and oil cut can be obtained.
(24)$$\left\{\begin{array}{ll} K_{w}^{pre}=1-1.10165\left[Y_{o}+\frac{0.02998Y_{o}{\left(1-Y_{o}\right)}^{2.7}}{U_{m}^{pre}}\right] & \left(D\;\mathrm{OS}/W,\;\mathrm{Transition}\;\mathrm{Flow}\right)\\ K_{w}^{pre}=1-0.91257\left[Y_{o}+\frac{0.0256Y_{o}{\left(1-Y_{o}\right)}^{2.4}}{U_{m}^{pre}}\right] & \left(D\;O/W,\;\mathrm{VFD}\;O/W\right) \end{array}\right..$$

First, flow pattern is identified based on the high-speed camera image, and then the corresponding model is selected in Equation (24) for calculation. The results are shown in Figure 13. The prediction of water cut has an absolute average deviation (AAD) of 0.01 and an average absolute percentage deviation (AAPD) of 2.08%. For the oil cut prediction, the absolute average deviation (AAD) is 0.01 and the average absolute percentage deviation (AAPD) is 8.86%. Both the prediction results are satisfactory. As shown in Figure 13, large deviations happen in D OS/W, as the slippage effect is very serious, and the phase distribution is extremely uneven. As shown in Figure 8, the sensor response under these low flow rate conditions (0.0184 m/s and 0.0369 m/s) is nonlinear, which results in large deviations in the measurement results. If these conditions are not considered, the measurement results can be more accurate.

## 7. Conclusions

According to the characteristics of low fluid productivity and high water-cut in the development of oil wells at present, a plug-in conductance sensor array was designed to measure the flow parameters of oil-water two-phase flow at low flow rate. By placing the insert body in the pipe, the fluid flow space is reduced, the flow velocity is increased, and therefore, both the slippage effect and the non-uniform distribution of dispersed phase are reduced. The geometries of the sensor are determined by finite element analysis. The sensor measurement performance of the water holdup and cross-correlation velocity was investigated by dynamic experiments. Prediction of mixture velocity and phase volume fraction are realized based on kinematic wave theory and drift flux model. The conclusions are as follows:
A plug-in conductance sensor array used for measurement of oil-water two-phase flow at low flow rate is proposed. The sensor is optimized by finite element analysis method, and the geometrical dimensions of the electrodes are determined according to the average sensitivity and the sensitivity variation parameter. Optimization simulation results show that when the height of the electrode is 2 mm, the distance between the two electrodes is 7 mm, and the distance between the upstream and downstream sensors is 20 mm, the performance of the plug-in conductance sensor is the best.Dynamic experiment is carried out on the oil-water two-phase flow experimental facility. The results show that the plug-in conductance sensor still has high resolution under high water-cut conditions, and the measured values show a significant step response with small changes in water cut. The combination of upstream and downstream sensors has good measurement characteristics of cross-correlation velocity. Under the same flow pattern, cross-correlation velocity *U_cc_* and mixture velocity *U_m_* have a good linear relationship.Considering the slippage effect and phase distribution characteristics of oil-water two-phase flow with low flow rate and high water cut, the mixture velocity measurement model and drift flux model are established, and the mixture velocity and phase volume fraction are predicted based on the measurement results of water holdup and cross-correlation velocity. The results show that the measurement results have high accuracy, and the AAD and AAPD of mixture velocity are 0.01 m/s and 8.5%, respectively. The AAD and AAPD of water cut are 0.01 and 2.08%, respectively. And the AAD and AAPD of oil cut are 0.01 and 8.86%, respectively.

## Figures and Tables

**Figure 1 sensors-19-04649-f001:**
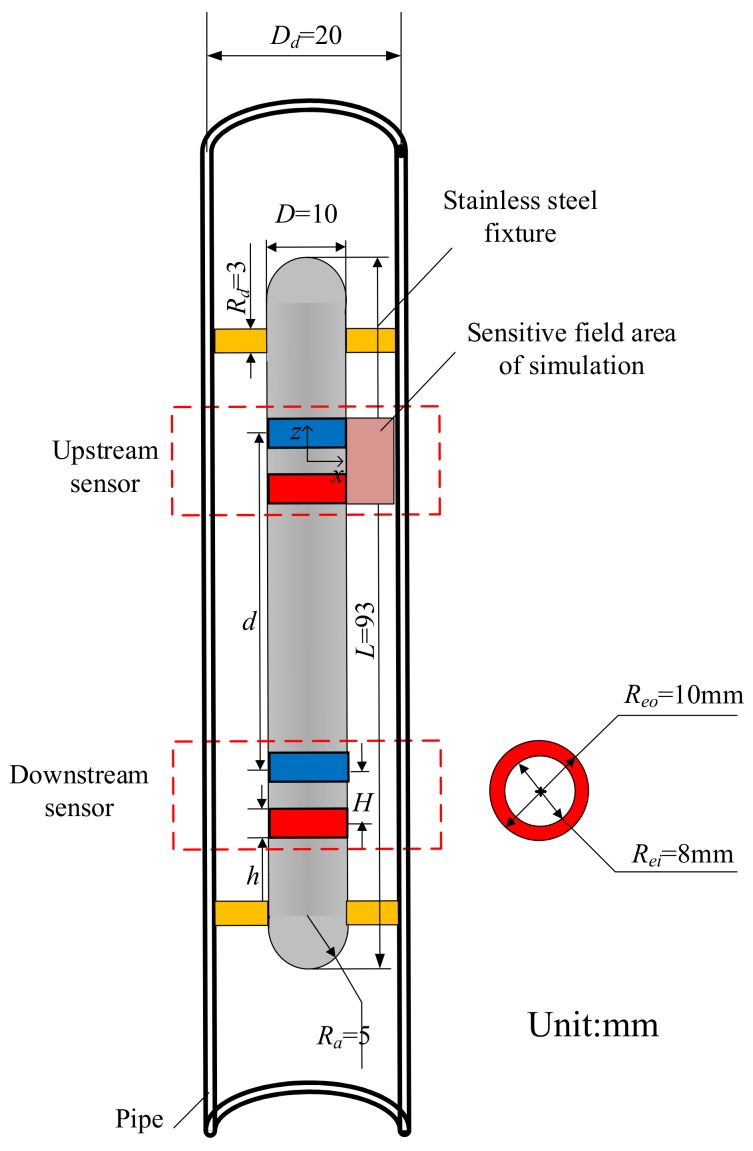
Structure of the plug-in conductance sensor array.

**Figure 2 sensors-19-04649-f002:**
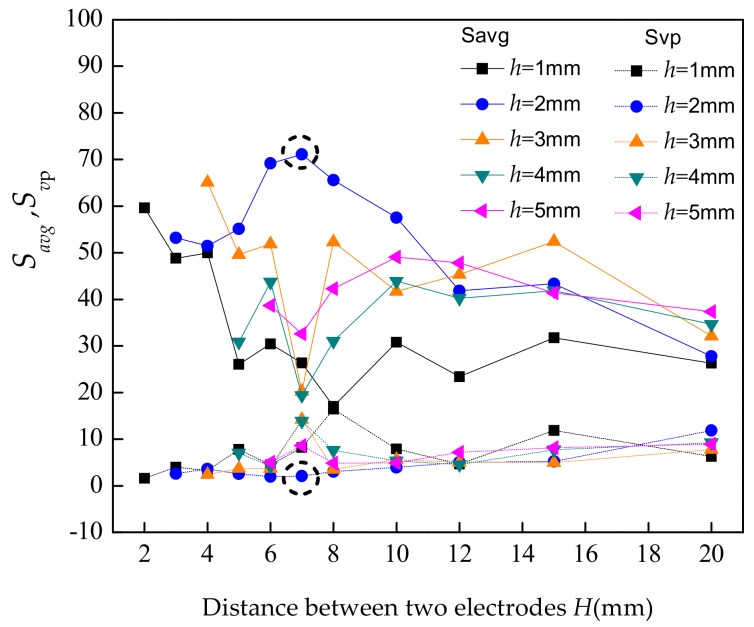
S_avg_ and *S_vp_* of sensitive fields under different sensor configurations.

**Figure 3 sensors-19-04649-f003:**
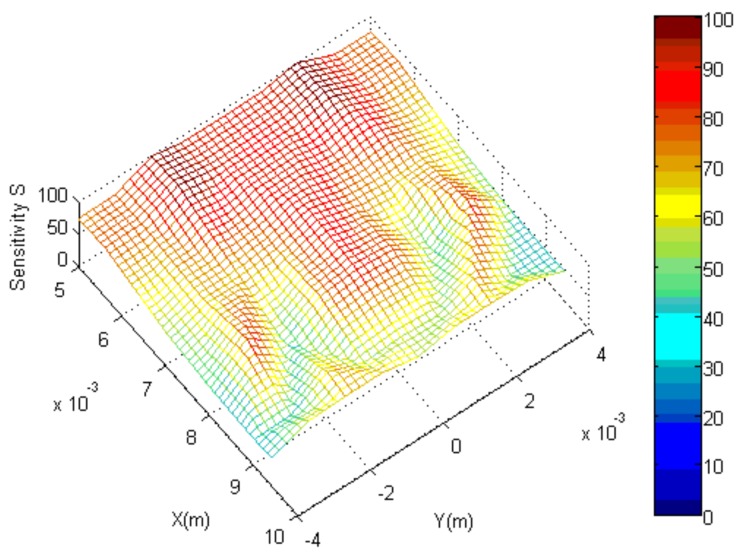
Distribution of sensitive field when the distance *H* = 7 mm and the height of the conductance ring *h* = 2 mm.

**Figure 4 sensors-19-04649-f004:**
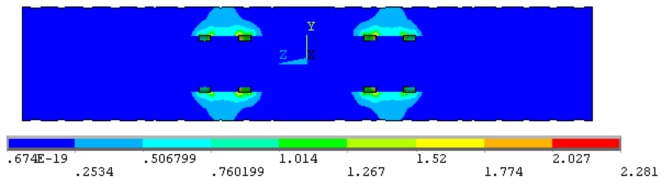
The electric field distribution when the distance *d* between the upstream and downstream sensors is equals to 20 mm.

**Figure 5 sensors-19-04649-f005:**
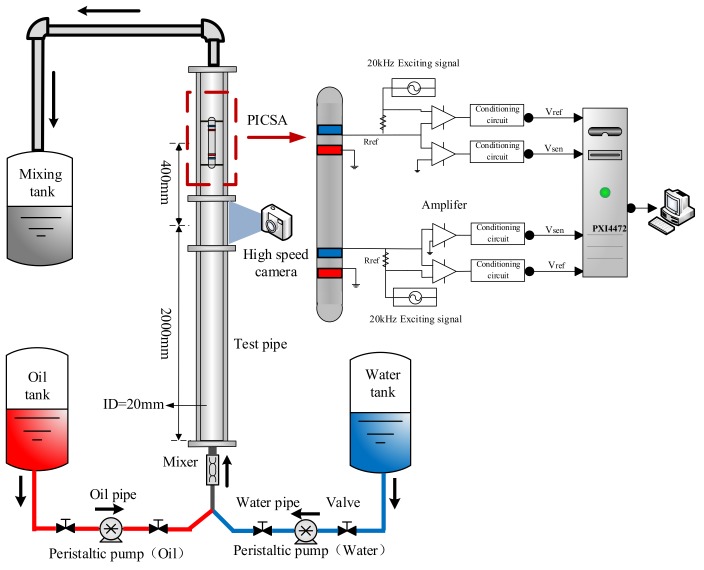
Schematic diagram of oil-water two-phase flow experimental facility.

**Figure 6 sensors-19-04649-f006:**
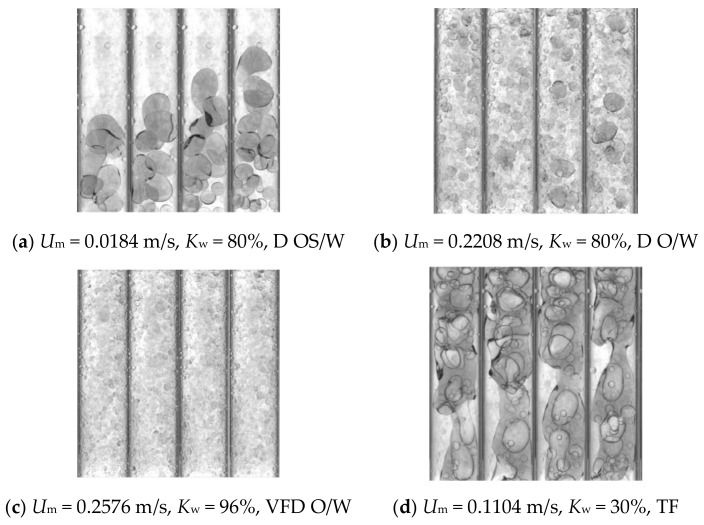
Snapshots of experimental flow patterns captured by a high-speed camera.

**Figure 7 sensors-19-04649-f007:**
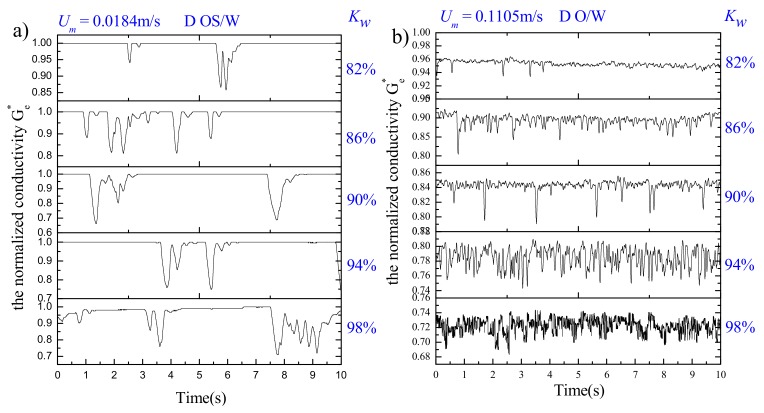
The normalized conductivity with different flow conditions. (**a**) D OS/W; (**b**) D O/W; (**c**) VFD O/W.

**Figure 8 sensors-19-04649-f008:**
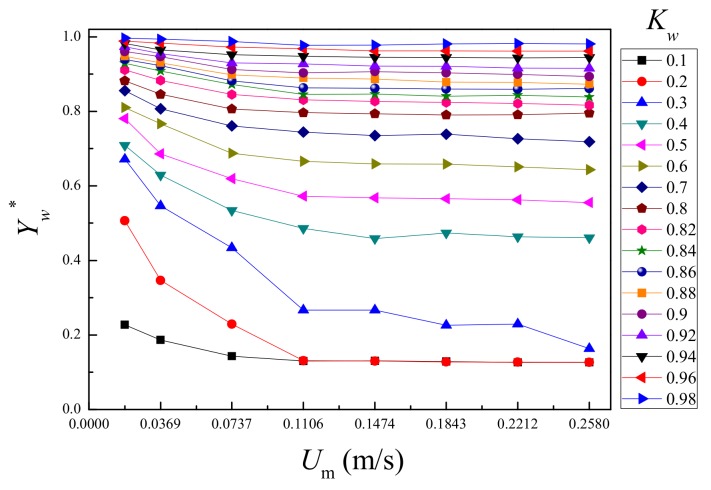
The response of the plug-in conductance sensor array in oil-in-water flows.

**Figure 9 sensors-19-04649-f009:**
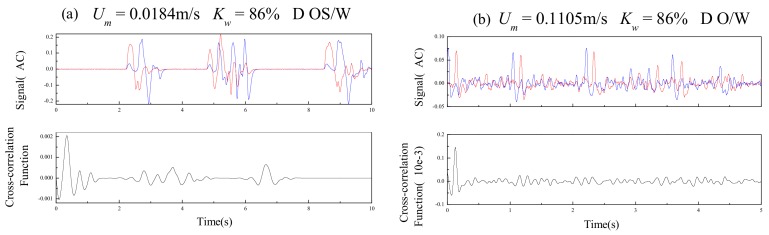
Cross-correlation velocity signal measured by plug-in conductance sensor.

**Figure 10 sensors-19-04649-f010:**
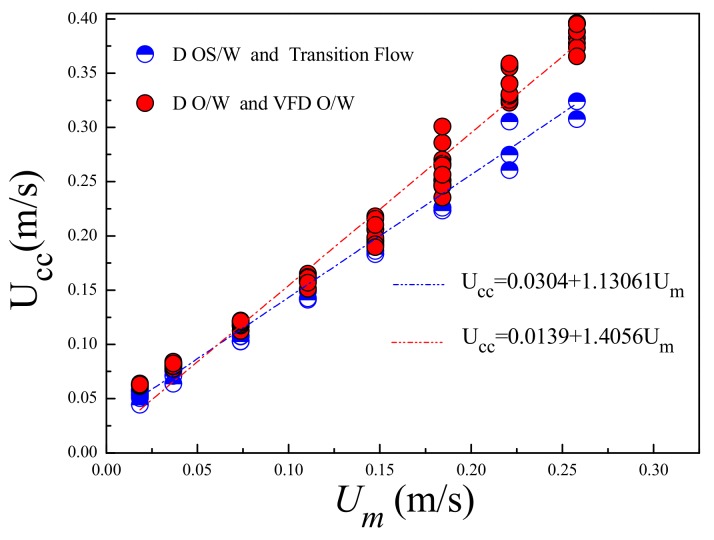
Relationship between cross-correlation velocity *U_cc_* and mixture velocity *U_m._*

**Figure 11 sensors-19-04649-f011:**
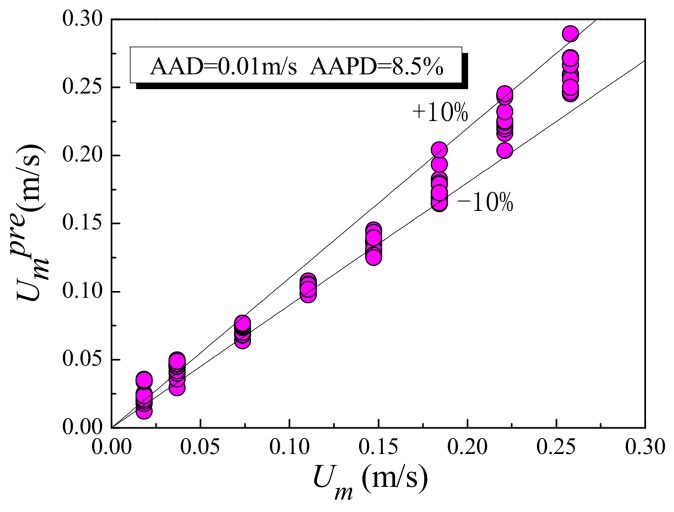
Relationship between predicted velocity *U_m_**^pre^* and mixture velocity *U_m_*.

**Figure 12 sensors-19-04649-f012:**
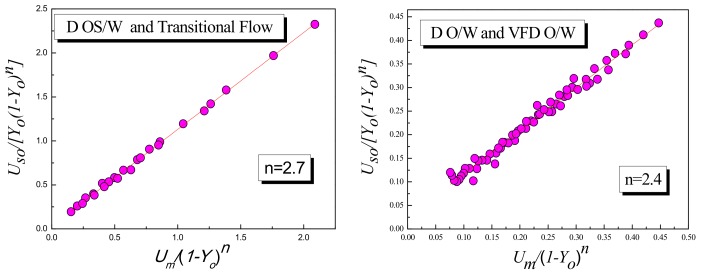
Relationship between $U_{m}/{\left(1-Y_{o}\right)}^{n}$ and $U_{so}/Y_{o}{\left(1-Y_{o}\right)}^{n}$.

**Figure 13 sensors-19-04649-f013:**
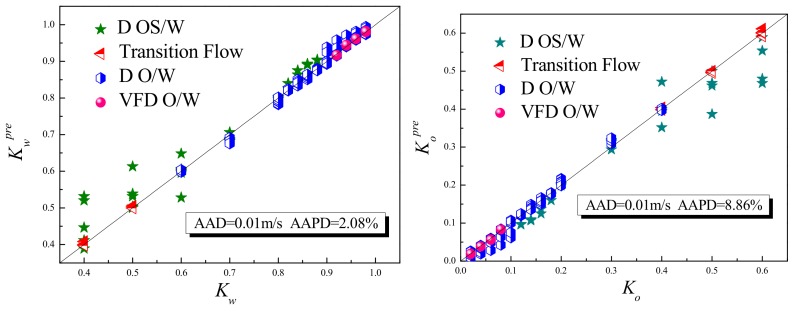
Prediction results of water cut and oil cut.

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
