# Peer review of "Flow Measurement of Oil-Water Two-Phase Flow at Low Flow Rate Using the Plug-in Conductance Sensor Array"

_sensors, 2019, doi:10.3390/s19214649_

Round 1
Reviewer 1 Report
Dear Authors.
The study you performed is very interesting and prepared in professional way. It is clear and reasonable, and contains all necessary sections. Using fast-frame camera for registration of the flow seems to be particularly interesting solution, which implements a bit of reality to the mathematical reasoning presented. I have no doubts that the study is suitable for publication in excellent scientific journal, which is Sensors. I suggest only one minor correction of a mistake noticed:
Page 1, line 38-39: you probably missed the reference [6].
Sincerely
Author Response
Response to reviewer 1 reports
Reviewer1: The study you performed is very interesting and prepared in professional way. It is clear and reasonable, and contains all necessary sections. Using fast-frame camera for registration of the flow seems to be particularly interesting solution, which implements a bit of reality to the mathematical reasoning presented. I have no doubts that the study is suitable for publication in excellent scientific journal, which is Sensors. I suggest only one minor correction of a mistake noticed:
Page 1, line 38-39: you probably missed the reference [6].
Response: Thanks for your valuable comments. We carefully checked the reference labels, [6] is a wrong reference, we have deleted it in the list.

Reviewer 2 Report
The paper presents an interesting technology in an application in need of such a sensor. The manuscript is well-organized, however, a few important comments are listed below:
It is hard to justify whether a flow rate is "high" or "low" without putting it into perspective. This especially is the case when you are using a relative big range between 0.0184-0.2580 m/s. The same goes for the water cut. Please address this as early as possible in the paper, e.g. by coming with real industrial examples Although the state-of-the-art is well described in the introduction, it is relevant to mention in which applications it can be used. It is espcially important when talking about which operating conditions are worth considering. Some previous studies have actually experimentally investigated a very similar approach for [1] Oil-in-Water 2-phase flow around de-oiling hydrocyclones as part of produced water treatment and [2] for 3-phase slug flow where the water cut is ~90% like at mature wells. Other similar studies probably exist too. In line 164 the sampling frequency is listed to be 4kHz, which seems to be extremely high compared to what is needed. Please discuss this. Furthermore, there also lack argumentation why sampling time for each flow condition is picked to be 30 s and 2 min, respectively (see line 164-165). Figure 5 is nice and illustrative, but also confusing with the current (lack of) description. Please modify to give the reader a better overview, e.g. by adding a table explaining all instrumentation with measurement/actuator range, units and measurement/actuator uncertainties. Figure 7 is too small to read in the current form. Enlarge the sub-figures. On a side note: It is nice to see vector graphic is used which obviously helps the reader in the papers digital form. Consider adding redundancy tests to make a statistical analysis on the experiments. In the current form the variations from test to test is not examined sufficiently. This is important in order to reduce/justify the effect of potential operational variations and outside disturbances. In fact, this should be a common practice for all experiments in any scientific study.
[1] Cost-Effective ERT Technique for Oil-in-Water Measurement for Offshore Hydrocyclone Installations, https://www.sciencedirect.com/science/article/pii/S2405896315008903
[2] Online Slug Detection in Multi-phase Transportation Pipelines Using Electrical Tomography, https://www.sciencedirect.com/science/article/pii/S2405896315008927
Author Response
Response to reviewer 2 reports
Reviewer 2: The paper presents an interesting technology in an application in need of such a sensor. The manuscript is well-organized; however, a few important comments are listed below:
Point 1: It is hard to justify whether a flow rate is "high" or "low" without putting it into perspective. This especially is the case when you are using a relatively big range between 0.0184-0.2580 m/s. The same goes for the water cut. Please address this as early as possible in the paper, e.g. by coming with real industrial examples
Response 1: Thanks for your valuable comments. At present, the water cut of old oil fields is generally over 80% in china. For example, at the end of 2007, the comprehensive average water cut of all new and mature oil fields owned by the three major oil companies in China has reached 86.0%. Generally, when the water cut exceeds 80%, oilfields are recognized to enter the late stage of high water-cut development [1]. When the flowrate is less than 0.2580 m/s, the slippage effect between phases is extremely seriously and the distribution of dispersed phase is extremely non-uniform than those at high flow rate conditions. These flow conditions are defined as low flow rate conditions. Under the low flow rate and high water-cut conditions, the flow measurement is a great challenge, and it is urgently needed to be investigated. Therefore, this paper limits the flow conditions to the stage of low flow rate and high water cut. We have added the necessary additions to the text (see line 34-42).
Point 2: Although the state-of-the-art is well described in the introduction, it is relevant to mention in which applications it can be used. It is especially important when talking about which operating conditions are worth considering. Some previous studies have actually experimentally investigated a very similar approach for [1] Oil-in-Water 2-phase flow around de-oiling hydro cyclones as part of produced water treatment and [2] for 3-phase slug flow where the water cut is ~90% like at mature wells. Other similar studies probably exist too.
Response 2: Thanks for your valuable comments. The applications it can be used is in oil-water two-phase flow at low flow rate and high water cut. It is especially important to design sensors with high resolution under high water-cut conditions. We have carefully read the two articles provided[1,2], the ERT method has a great performance in this situation, thus we quoted the two articles in the introduction to complete our study. (see line 84-88).
Point 3: In line 164 the sampling frequency is listed to be 4 kHz, which seems to be extremely high compared to what is needed. Please discuss this.
Response 3: Thanks for your valuable comments. The output signals reflect the characteristic of flow whose frequency is generally lower than 50 Hz as presented in our previous study [3]. In theory, the higher the sampling frequency is, the more accurate the obtained flow information is, but considering the amount of calculation data, it is reasonable to set the sampling frequency as 4 kHz.
Point 4: Furthermore, there also lack argumentation why sampling time for each flow condition is picked to be 30 s and 2 min, respectively (see line 164-165).
Response 4: Thanks for your valuable comments. When the flow pattern is dispersed oil-in-water slug flow, two-phase flow rate is very low at this time, and it will take a long time for the oil slug to flow through the sensor. Sampling time is too short to acquire complete flow pattern information. To ensure the integrity and repeatability of the flow structure (see Response 6 for a discussion of repeatability), the sampling time is chosen to be 2 min. As flow rate increases, the oil plug is broken into smaller oil bubbles and further dispersed into very fine oil bubbles. the 30s sampling time is enough to obtain complete flow information in these situations.
Point 5: Figure 5 is nice and illustrative, but also confusing with the current (lack of) description. Please modify to give the reader a better overview, e.g. by adding a table explaining all instrumentation with measurement/actuator range, units and measurement/actuator uncertainties.
Response 5: Thanks for your valuable comments. The test section is an acrylic material pipe with 20 mm inner diameter. In order to acquire the fully developed flow patterns in the pipe, a high speed camera is installed at the height of 2000 mm from the entrance to record the flow structures. Afterward, the plug-in conductance sensor array is mounted at the height of 400 mm from the camera. So that the single sensor measures the water holdup, and the two upstream and downstream sensors measure cross-correlation velocity. In addition, experimental flow mediums include tap water and No. 3 industrial white oil stored in water and oil tank respectively and separated in the mixing tank. Two industrial peristaltic pumps are used to transport and control the flow of water and oil, respectively, with an uncertainty of ±0.2%[4]. The peristaltic pump working based on the discrepancies of rotating speed should be calibrated before the experiment when fluid mediums have distinctive physical properties, such as density and viscosity. Specific calibration procedure can refer to our previous study [4]
The analog sinusoidal signals with 20 kHz frequency and peak to peak value of 4 V are applied to the exciting electrodes through reference resistant Rref, then the measuring electrodes are connected to the ground. The voltages in the sensors and reference resistant Rref are demodulated and the outputs of conductance sensors are collected with PXI4472 synchronous acquisition card produced by NI Company. According to the research results reported by Han et al. [4], the highest flowing frequency of low-velocity oil–water flows is about 50 Hz. Herein, 4 kHz is selected as suitable sampling frequency. We have added the necessary additions to the text (see line 168-179 / 182-188).
Point 6: Figure 7 is too small to read in the current form. Enlarge the sub-figures. On a side note: It is nice to see vector graphic is used which obviously helps the reader in the papers digital form. Consider adding redundancy tests to make a statistical analysis on the experiments. In the current form the variations from test to test is not examined sufficiently. This is important in order to reduce/justify the effect of potential operational variations and outside disturbances. In fact, this should be a common practice for all experiments in any scientific study.
Response 6: Thanks for your valuable comments. Figure 7 has been modified seen line 234-235. The fluctuating signals of PICSA under different flow conditions are shown in Fig.7. For each flow condition, the fluctuating signals of PICSA have stable base value within the sampling time, and the signal fluctuation characteristics have good repeatability with time. All those characteristics can indicate that the PICSA possess high repeatability of measurement.
[1] Cost-Effective ERT Technique for Oil-in-Water Measurement for Offshore Hydrocyclone Installations, https://www.sciencedirect.com/science/article/pii/S2405896315008903
[2] Online Slug Detection in Multi-phase Transportation Pipelines Using Electrical Tomography, https://www.sciencedirect.com/science/article/pii/S2405896315008927
[3] A Characterising Dynamic Instability in High Water-Cut Oil-Water Flows Using High-Resolution Microwave Sensor Signals. Liu W X et al. 2018 Zeitschrift Für Naturforschung, https://doi.org/10.1515/zna-2018-0003
[4] Differential pressure method for measuring water holdup of oil–water two-phase flow with low velocity and high water-cut. Y.F. Han et al.2016, https://doi.org/10.1016/j.expthermflusci.2015.11.008
